# Geo-Environmental Assessment of Tourist Development and Its Impact on Sustainability

Fernando Morante-Carballo [1,2,3], Boris Apolo-Masache [1,2,4,*], Fred Taranto-Moreira [5], Bethy Merchán-Sanmartín [1,3,6], Lady Soto-Navarrete [7,8], Gricelda Herrera-Franco [3,9] and Paúl Carrión-Mero [1,6,*]

1 Centro de Investigaciones y Proyectos Aplicados a las Ciencias de la Tierra (CIPAT), ESPOL Polytechnic University, Campus Gustavo Galindo Km 30.5 Vía Perimetral, Guayaquil P.O. Box 09-01-5863, Ecuador
2 Facultad de Ciencias Naturales y Matemáticas (FCNM), ESPOL Polytechnic University, Campus Gustavo Galindo Km 30.5 Vía Perimetral, Guayaquil P.O. Box 09-01-5863, Ecuador
3 Geo-Recursos y Aplicaciones (GIGA), ESPOL Polytechnic University, Campus Gustavo Galindo Km. 30.5 Vía Perimetral, Guayaquil P.O. Box 09-01-5863, Ecuador
4 Centro Del Agua y Desarrollo Sustentable (CADS), ESPOL Polytechnic University, Campus Gustavo Galindo Km 30.5 Vía Perimetral, Guayaquil P.O. Box 09-01-5863, Ecuador
5 Facultad de Ciencias Ambientales, Campus Ing. Manuel Haz Álvarez, Universidad Técnica Estatal de Quevedo, Av. Quito km. 1 1/2 Vía a Santo Domingo de los Tsáchilas, Quevedo P.O. Box 120301, Ecuador
6 Facultad de Ingeniería en Ciencias de la Tierra (FICT), ESPOL Polytechnic University, Campus Gustavo Galindo Km 30.5 Vía Perimetral, Guayaquil P.O. Box 09-01-5863, Ecuador
7 Facultad de Ciencias Sociales y Humanísticas (FCSH), ESPOL Polytechnic University, Campus Gustavo Galindo Km. 30.5 Vía Perimetral, Guayaquil P.O. Box 09-01-5863, Ecuador
8 Grupo de Investigación en Turismo Marino y Costero, ESPOL Polytechnic University, Campus Gustavo Galindo Km. 30.5 Vía Perimetral, Guayaquil P.O. Box 09-01-5863, Ecuador
9 Facultad de Ciencias de la Ingeniería, Universidad Estatal Península de Santa Elena (UPSE), Avda. Principal La Libertad–Santa Elena, La Libertad P.O. Box 240204, Ecuador
* Correspondence: bhapolo@espol.edu.ec (B.A.-M.); pcarrion@espol.edu.ec (P.C.-M.);
Tel.: +59-3985641396 (B.A.-M.); +59-3998265290 (P.C.-M.)

**Abstract:** The evaluation of geosites is an essential part of conserving the geodiversity and biodiversity of an ecosystem, as well as safeguarding the cultural, geological, environmental, and landscape wealth that a highly recognized geographical area possesses. In this context, Guayaquil, the pearl of the Pacific, is a city that registers, in its history and evolution, a binding relationship with the geo-biodiversity of the geosites that characterize it. This work aims to assess places of tourist interest in Guayaquil and its surrounding areas through a geo-environmental evaluation matrix to establish a sustainability proposal that promotes the geotourism development of the city. The methodology consisted of: (i) geographic registration of the chosen sites and present characteristics, (ii) environmental analysis of the sites based on the cause–effect method, and (iii) strategies for the inclusion of these sites into the geo-environmental and geotourism system of Guayaquil as potential geosites. The results demonstrate that Guayaquil has impressive geodiversity in several potential natural sites, obtaining highly representative values that reinforce the city's natural diversity elements. Sites such as Cerro San Pedro and the Cerro San Eduardo, Pascuales, and Zeolites quarries have the most significant adverse environmental impacts. While places such as Cerro Azul, Estero Salado, Isla Santay, and Hornos de Cal (Bosque Protector Cerro Blanco), obtained positive values that highlight their environmental value, being of great benefit to the city and to nature. Furthermore, some of these places could be integrated into tourism development plans, and as potential geosites, they could complement various services and opportunities for discovering nature. Finally, all this can lead to a sustainable proposal for a Global Geopark project in Guayaquil based on the results obtained in this work.

**Keywords:** cause–effect matrix; environmental impact; geosites; conservation; ecotourism; natural heritage

## 1. Introduction

The most outstanding scenic and unique places on planet Earth are undoubtedly a sample of geological evolution [1] and the interaction between the environment and living beings [2,3]. The characteristics present in each site and the participation of the local community make these megadiverse places, where geodiversity and biodiversity abound which make for a potentially touristic geological site [4–6]. Places such as Toba Caldera Geoheritage (Sumatra island, Indonesia) [7], the bio-geocultural areas of the Mediterranean region of Chile [8], the humid mountains in the northeast of Brazil [9], mining sites [10], and in general, geotrails or places with geological and mining heritage [11–15].

After the Convention on Biological Diversity (in 1992), the term "geodiversity" arose in 1993 among geoscientists, who proposed this term at the end of the 1990s until the International Union for Conservation of Nature finally recognized it [16,17]. Geodiversity is a set of sites with unique natural characteristics, made up of geological, geomorphological, and landscape or soil systems created by biological processes or human activity [18,19]. According to Gray M. [5], this term is the abiotic equivalent of biodiversity, providing goods and services to society, it is essential as a backbone for geoheritage and geoconservation (elements of geodiversity worth conserving). Contrary to this, biodiversity focuses on the relationship between the flora and fauna of a sector and the geological history they present [20,21].

Evaluating the geodiversity of a geological and/or mining site (geosite) can greatly support the designation of future Geoparks [22,23]. These areas highlight the most important aspects of the geographical sector (nationally and internationally), safeguarding geological and landscape wealth through strategies that contribute to the conservation of the environment [24–26] and education in earth sciences and promotion of sustainable development of the local economy [27,28]. In this context, geotourism plays an important role in the development of Geoparks [29–32]. In addition, 177 geoparks exist scattered throughout the 46 countries with official recognition, directly endorsed and monitored by the United Nations Educational, Scientific and Cultural Organization (UNESCO) [33]. The Geopark concept was conceived of as a tool for the protection and promotion of the geological heritage of the Earth, it was discussed for the first time at the Digne Convention in 1991 and obtained the support of UNESCO in the Global Geoparks Program [34].

The unique aspects or characteristics of the geosites promote the sector's scientific, educational, and tourist interests [35,36]. Moreover, in this way, considering the social, historical, and cultural value of geoheritage [37–40], can influence socio-economic development [41] and sociocultural sustainability [42]. In this regard, implementing geotourism in areas with natural and cultural wealth can significantly contribute to the geoconservation of geosites [43–46]. These present a patrimonial value (cultural, geological, or mining) inside or outside a city that deserves to be recognized [47–50].

Geoconservation is the set of actions aimed at preserving the geological heritage and existing biodiversity in a geosite [51–53] through using natural resources in a socially just and economically viable manner [54,55]. In recent decades, in some places, with the support of local participation, geoconservation and the evaluation of geosites have been promoted [56–58], allowing for to registration and management of these places [59–61]. One example is Jeju Island in South Korea. Thanks to its geodiversity, biodiversity, and socio-cultural aspects, it has achieved several nominations [62,63] as a natural heritage site of humanity, a biosphere reserve, a Global Geopark, and a RAMSAR wetland. As a result, it has increased scope and objectives [52], whereas environmental geology as a study focused on the interaction of human activities with the environment, hazards, and the natural resources of the Earth [44]. IUCN and several researchers [64] highlight the importance of conserving the integrity and natural diversity of the earth through guidelines and adequate management of the areas for the long-term conservation of nature.

Initially, the environmental impact assessment was conducted using matrices, such as Leopold's, which considered the actions and environmental factors in an ecosystem [65,66]. This evaluation is a useful procedure or tool to describe the environmental impacts of

projects, works, or human activities to be accepted, modified, or rejected by the respective entity that regulates them [67]. The cause–effect matrix is an evaluation method derived from the Leopold matrix [68], an improved and adaptable version that makes it possible to identify and evaluate environmental performance more effectively, analyzing the cost-benefit of social activities and the flow of inputs, products, and waste generated [69].

Among the environmental impacts of this reality, the profound transformation of the land and outdoor activities stand out, such as geological and natural reserves that can be sustainably used in environmental tourism and geotourism [65,70]. In addition, exploring and identifying environmental problems in geosites promote the care and mitigation of risks that can affect society and the environment [71,72]. The conservation of these sites as protected areas is a process that would generate economic income in the region, creating benefits for local communities and encouraging the application of environmentally friendly geotourism [54,73,74].

The Sustainable Development Goals (SDGs) adopted by the United Nations in 2015 contain the most ambitious global agenda approved by the international community [75]. Among its objectives are various key aspects such as climate change, environmental degradation, developing inclusive and sustainable cities, geoheritage, geotourism, and geoeducation [76,77]. The UNESCO Global Geoparks (UGGs) and the contributions it provides are focused on the direct assistance of eight SDGs (1, 4, 5, 8, 11, 12, 13, and 17) [78,79]. However, directly or indirectly, it has been claimed that they contribute to the 17 established SDGs [80], which could greatly benefit the different key aspects present in geosites or sites of interest.

Guayaquil is one of the most populated and largest cities in Ecuador. It presents a tourism potential linked to commercial and recreational activity in its geosites due to its landscape characteristics and available outdoor activities [81,82]. In addition, given the confinement related to the COVID-19 pandemic, people are seeking spacious, natural, and diverse places, so the development of many places should be increased, aimed at creating a sustainable city and tourist destination [83]. The implementation and promotion of geotourism in Guayaquil meet various SDGs through tourism development in natural areas [84,85]. These focus specifically on geo-biodiversity and the heritage value of the place, allowing it to become an instrument to promote economic development, geoconservation, and geoeducation about its natural heritage [86–88].

The study city has several sites of geological and mining interest, in which geodiversity linked to the three geomorphological macro-domains of Guayaquil stands out (see Figure 1): (i) the alluvial plain of the Daule and Babahoyo rivers, (ii) the estuarine-deltaic plain of the Guayas river, and (iii) the Chongón-Colonche mountain range. Furthermore, geologically, Mesozoic and Cenozoic rocks, Quaternary sedimentary deposits, and volcanic sites stand out in the city and its surroundings [89,90]. Some sites are formed by natural fluvial/intertidal processes, rock outcrops with landscape features, and quarries due to their geological composition. There is a unique local geodiversity defining the geoheritage of the area linked to natural, ancient, and historical areas framed by the natural environment, urban development, and human activities of yesteryear.

Guayaquil presents sites with a high tourist, scientific, and educational index, influencing the possibility of sustainable development for their environment and the community [90]. Therefore, what strategic elements are considered to establish a sustainability proposal through a geo-environmental evaluation? This work aims to assess eight sites of geological and environmental interest, in the tourist context of Guayaquil and its surrounding areas, through a geo-environmental evaluation matrix that considers the different aspects of impacts using a cause–effect analysis for a proposal of sustainability as an tool for decision-makers that promotes the development geotourism within the city.

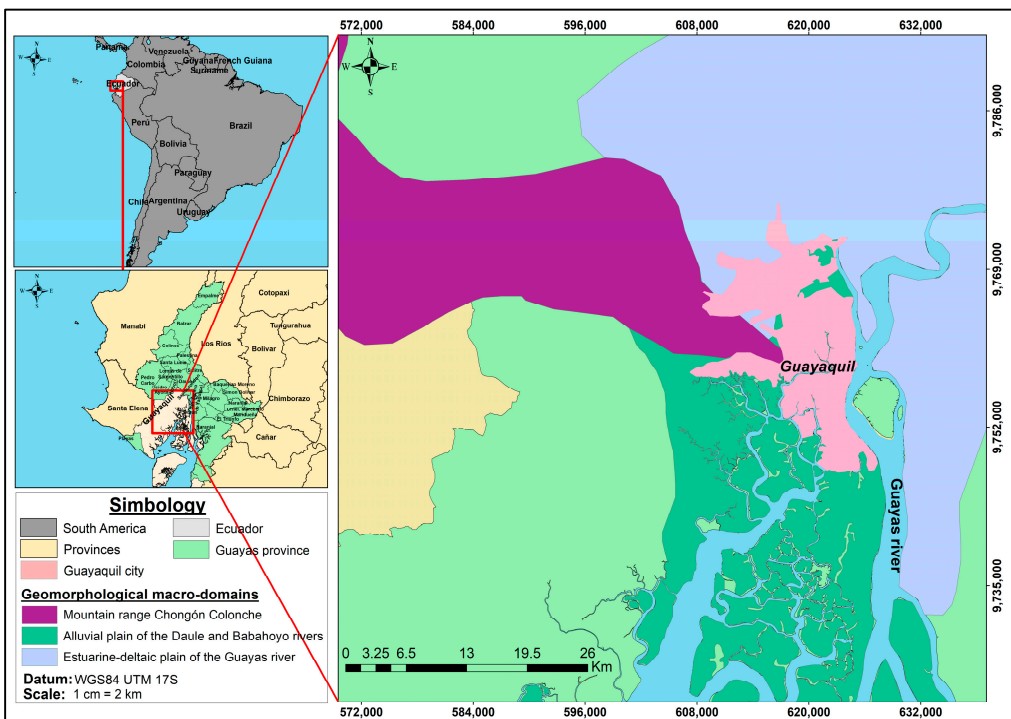

**Figure 1.** Geographical and geomorphological map of the study area. Source: adapted from [91].

## 2. Materials and Methods

The following study has three phases (Figure 2): (i) geographic record of the chosen geosites and present characteristics, (ii) environmental analysis of the geosites based on the cause–effect method, and (iii) strategies for inclusion of these sites in the Guayaquil geo-environmental and geotourism system, to promote sustainable development in the city.

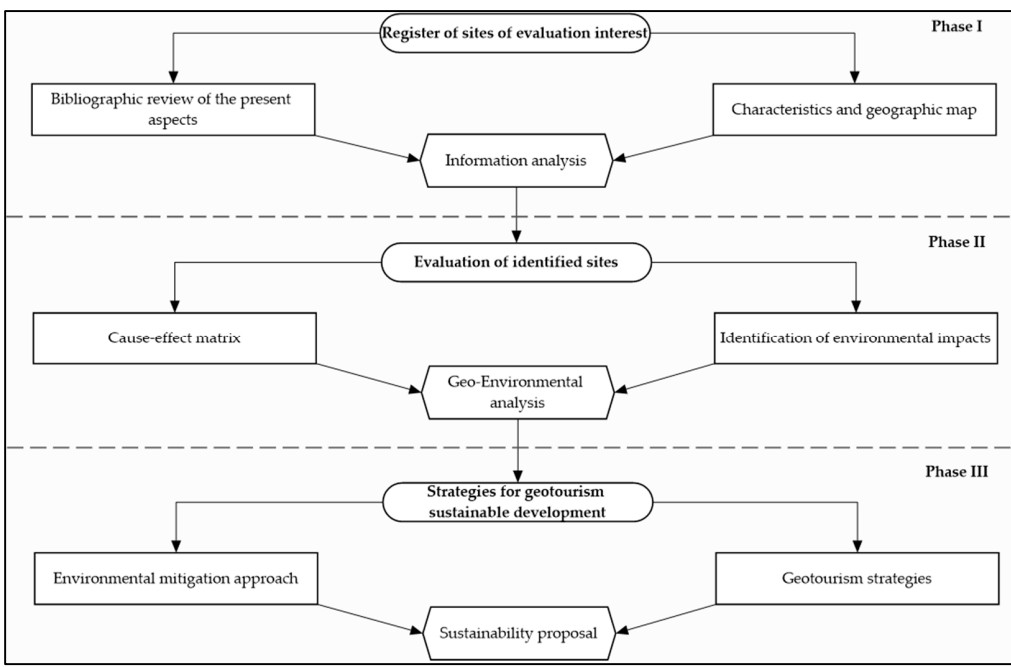

**Figure 2.** Workflow of the present study.

*2.1. Geographical Registration and Characteristics of Sites of Interest*

This phase consisted of a bibliographic review of the sites of interest around Guayaquil to determine their main characteristics from a previous study [90], in which they valued 12 places through the Brilha-Medina (2015) methodology [92], focused on the tourist, educational, and research aspects. As a result, these places have more than 50% approval as potential geosites (see Table 1). However, a subsequent study evaluated the geo-environmental impact at three locations from the same article [93].

**Table 1.** Quantitative evaluation of potential geosites. Obtained from: [90].

| Nº | Geological Places of Interest (LIGs, Acronym in Spanish) | Approval Percentage | Geosite Relevance Value | Scientific/Educational Use Value | Tourist Use Value | Geoconservation Index |
|---|---|---|---|---|---|---|
| 1 | Cerro Azul (hill) | 80.30% | 12.75 | 2.61 | 2.26 | 82.36% |
| 2 | Malecón del Estero Salado (estuary) | 74.24% | 11.42 | 2.36 | 2.22 | 74.44% |
| 3 | Isla Santay (island) | 69.70% | 11.08 | 2.33 | 1.90 | 69.70% |
| 4 | Horno de Cal (kilns) | 65.15% | 10.08 | 1.69 | 2.16 | 65.15% |
| 5 | Canteras de Pascuales (quarries) | 56.06% | 9.00 | 2.03 | 1.42 | 59.03% |
| 6 | Cantera de Zeolitas (Zeolite quarry) | 53.03% | 8.50 | 1.81 | 1.42 | 55.14% |
| 7 | Cerro San Pedro (hill) | 51.52% | 8.25 | 1.81 | 1.35 | 53.89% |
| 8 | Canteras del Cerro San Eduardo (quarries) | 50.00% | 8.33 | 1.89 | 1.13 | 54.31% |

Table 1 shows the percentage of approval of the geosites according to the relevance value (between 5.2 and 15.7), scientific/educational and tourism (both between 1 and 3), as well as the geoconservation index of each, since the higher the value, the better the conditions of each site. Additionally, because Puná Island is geographically far from the city, this work only considered the eight remaining sites:

These sites, as potential geosites, have a percentage of approval greater than 50%, optimal for the objective of this study since they are suitable for tourist and scientific/educational use of the geodiversity and geoheritage they represent. However, these places present various aspects with positive and negative influences (see the results section in Section 3.2). Therefore, through field visits, they were identified and evaluated from the experts' criteria.

The group of experts is made up of: Ph.D. Paul Carrión, Ph.D. Fernando Morante, Ph.D. Gricelda Herrera, MSc. Bethy Merchán, MSc. Lady Soto, Ing. Fred Taranto, and Ing. Boris Apolo (Table S1 of the supplementary material provides access to the academic profile of each evaluator). The experts addressed issues related to environmental aspects for their respective evaluation, which depended on the criteria/experience of the evaluator. Therefore, the impact value of the geosite would consider the average value assigned to each parameter.

*2.2. Environmental Analysis of the Sites Evaluated Based on the Cause–Effect Method*

The cause–effect matrix allows for an environmental impact assessment, observing the interactions and interrelationships between the different components of the natural and anthropogenic environment [94]. As shown in Figure 3, the parameters used are: "character" which indicates whether the identified aspect is positive or negative; "duration" and "reversibility" correspond to the time that the effect will remain, depending on its recovery capacity [93,95]. The parameters "probability", "intensity", and "extension" can be qualified in three values depending on the scale required by the evaluator [96]. The

parameter "importance" is related to the degree of relevance that varies from 1 to 10, with 10 being a very important interaction and 1 as a relatively unimportant interaction [97].

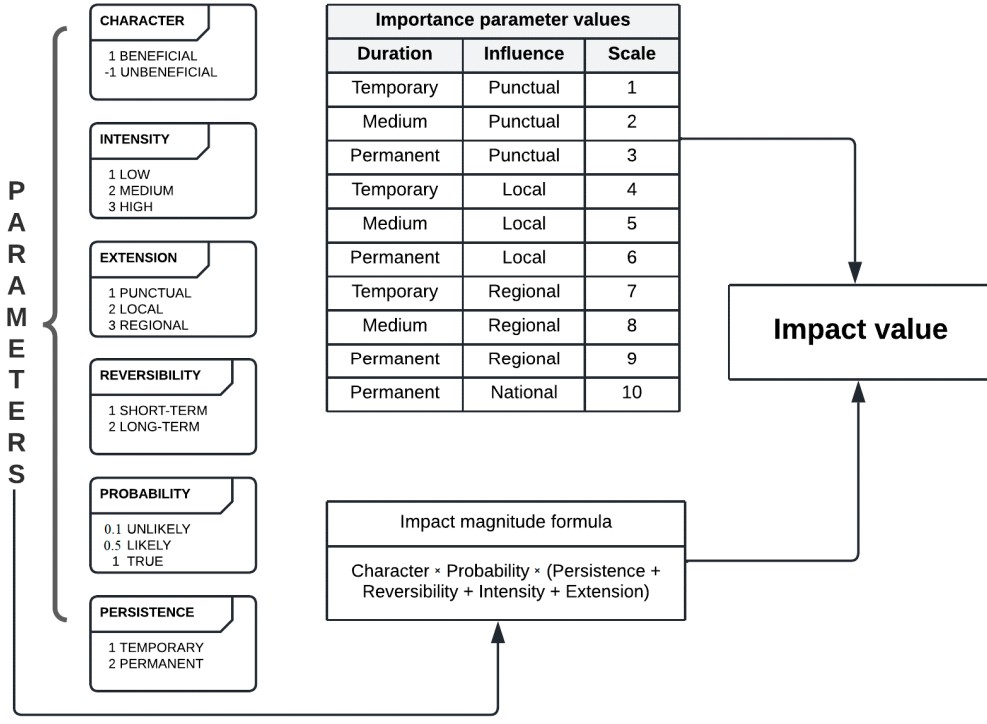

**Figure 3.** Diagram of the process for the assessment of environmental impact.

The identified factors make it possible to detect those aspects with information about the state of the environment. For its evaluation, the matrix has (i) rows with the environmental factors that may be affected and (ii) columns with the actions that may cause impact. This allowed for the detection of the positive and negative environmental consequences in each site of interest [98].

### 2.3. Analysis of Results and Strategies for Including Sites in a Geo-Environmental and Geotourism System

This phase analyses the results through the participation of expert researchers considering proposals and actions that could mitigate the identified environmental impact [99]. In addition, it focused on positive environmental aspects, contributed to implementing geotourism strategies that promote local development, and focused on the conservation and sustainable development of geosites in Guayaquil [100].

The negative values (−) results represent each evaluated site's adverse environmental impacts and the degree of magnitude. On the other hand, the positive aspects (+) identified through the cause–effect matrix are those aspects that benefit the sector in an environmentally friendly and sustainable way. Based on these data, it is feasible to make an environmental mitigation proposal, exposing solutions according to the strengths of each place [101,102]. In this context, it is possible to establish strategies linked to geotourism, geoconservation, geoeducation, and the mitigation of environmental impacts [103] in search of sustainability [41,85,104].

## 3. Results

### 3.1. Registration of Sites of Interest and Their Main Aspects Identified

The eight identified sites have various characteristics that highlight their tourist, scientific, and cultural interests. Four of these places are in the city's interior, while the rest are in the vicinity of the city's outskirts. The geographical map of Figure 4 shows the

location of each sector, while Table 2 presents a description of each valued site. In addition, this study established a strategic reference site for the geographical visualization of the evaluated areas. Cerro Santa Ana is the appropriate site due to its historical and tourist importance as well as its position in the Guayaquil environment.

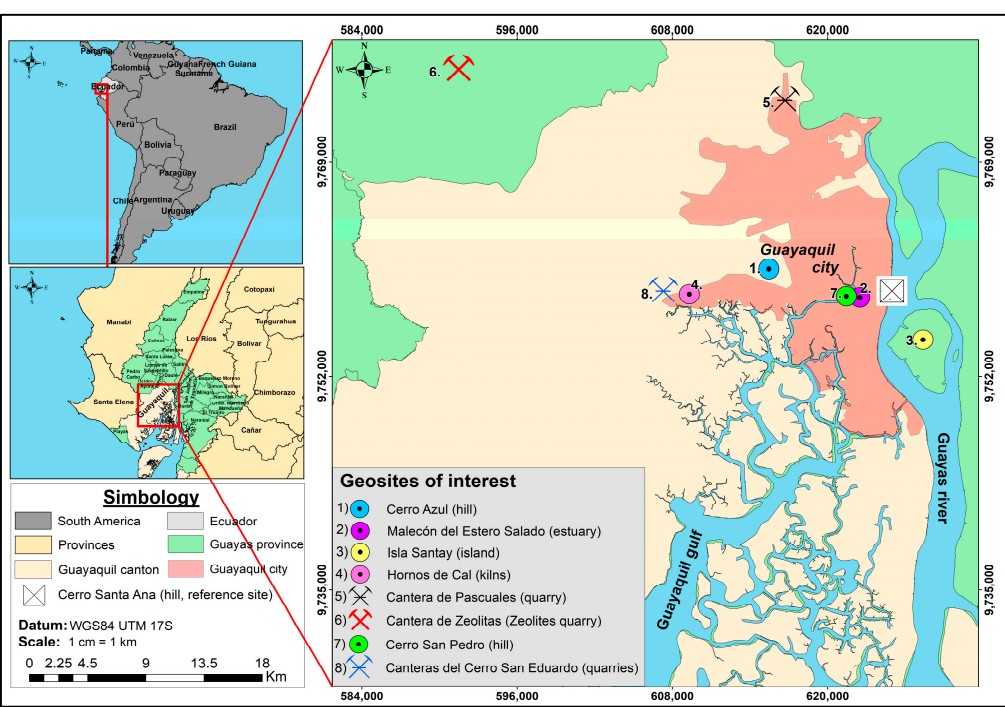

**Figure 4.** Geographical map of evaluation interest sites *. Source: adapted from [91]. * The numbers visible on the geographic map represent the name of each geosite in the box located at the bottom left.

**Table 2.** Registration of sites and their main features.

| Interest Site | Characteristics | Location: Santa Ana Hill Reference Point |
|---|---|---|
| Cerro Azul (hill) | It is an optimal place for recreational activities such as walking or cycling. It has a high relief and a bluish color where flora and fauna abound due to dense vegetation. Outcrops of volcano-sedimentary (Cayo Formation) and sedimentary (Guayaquil and San Eduardo Formations) rocks can be observed [103,104]. In addition, there are various trails where you can walk and see, in a panoramic way, the great expanse of Guayaquil and the road to the coast. | West (9–11 km) |
| Malecón del Estero Salado (estuary) | It has deltaic ramifications from the Pacific Ocean, generating a unique geomorphic and stratigraphic architecture in the sector [105,106]. It has a natural wealth represented by its mangroves and its biodiversity (flora and fauna). Much of the city lies on these deltaic-estuarine deposits. | West (3–3.5 km) |
| Isla Santay (island) | It is a protected natural area declared as a RAMSAR site and ecotourism wetland for social, cultural, and economic benefit [107,108]; it has clayey and sandy sediments deposited by the Guayas River and is influenced of the transgression and regression of the Pacific Ocean. It has a natural wealth that enhances social recreation, linking the flora and fauna that abound in the sector [109,110]. | Southeast (4–5 km) |

**Table 2.** *Cont.*

| Interest Site | Characteristics | Location: Santa Ana Hill Reference Point |
|---|---|---|
| Horno de Cal (kilns) | Located in the natural reserve of the Cerro Blanco Protected Forest and built with calcareous rocks from the sector as sources of lighting for night visits. This sector increases the link between the human being and nature through camps and picnics that improve the coexistence and comfort of visitors, managing to promote social and family recreation [111]. | West (16–18 km) |
| Canteras de Pascuales (quarries) | In this sector, there are several quarries on the Piñón Formation (Jurassic—Cretaceous), made up mainly of basaltic and diabasic rocks [89,112], where they extract rock material for the construction of roads and construction aggregates for the urban development of Guayaquil. | Northwest (17–20 km) |
| Cantera de Zeolitas (Zeolites quarry) | It has volcano-sedimentary rocks from the Cayo Formation (Upper Cretaceous) [113,114] and exploits zeolite rocks for different industrial and agricultural uses. In addition, there are characteristics of a tropical dry forest typical of the area. | Northwest (39–40 km) |
| Cerro San Pedro (hill) | It has a singular rocky stratification (almost perpendicular) and excellent stability for population settlement at the base of the hill. Furthermore, it is one of the hills belonging to the Cordillera Chongón-Colonche (Paleogene and Cretaceous rocks) and is mainly made up of silicified shales (Guayaquil Cherts) [89]. Therefore, they are ideal for study and research. | West (4–5 km) |
| Canteras del Cerro San Eduardo (quarries) | The hill has limestone rocks from the San Eduardo Formation (Upper Eocene) and alluvial soils with good mechanical resistance and low porosity [89], used for manufacturing and distributing cement throughout the country. In this sector, there are several companies dedicated to this type of exploitation. | West (13–15 km) |

*3.2. Environmental Aspects of the Guayaquil Geosites*

The identified environmental aspects are present in the current state of each place and identified through on-site visits. Table 3 shows the twenty-four aspects valued and categorized into four sections. For further details, Table S2 of the supplementary material shows each aspect described in 24 items for environmental assessment [115–143].

**Table 3.** Categorization of the twenty-four aspects identified in situ for environment assessment.

| Environmental Aspects | Cultural Aspects | Aspects with Political Management | Social Aspects |
|---|---|---|---|
| Gas emission perception | Symbol or figure cultural | Soil quality perception | Generation or presence of wastewater |
| Bad odor perception | | Electricity consumption | Industrial activity |
| Noise and vibration perception | | Construction for geosite adjustments | Commercial activities |
| Fauna | | Organic waste generation | Employment generation |
| Flora | | Inorganic waste generation | Service generation |
| Reforestation | | Hazardous waste generation | Social recreation |
| Loss of vegetation cover | | Presence of vehicles (transport or machinery) | Tourist capacity |
| Visual or landscape aspect | | | Tourist safety |

Figure 5 presents the impact values of the environmental aspect present in each geosite (differentiated by colors). The negative values (−) represent the adverse environmental impacts in the evaluated sites. On the other hand, the positive values (+) symbolize the strengths that provide sustainability at each site of analysis. In addition, the length of each bar is represented by the magnitude of the impact, for example: in the perception of gas emission, the first has a value of approximately 15, the second has a value of 7, the third a value of 12, the fourth a value of 30, the fifth a value of 50, the sixth a value of 7, the seventh a value of 10, the eighth a value of 40, and the ninth a value of 50.

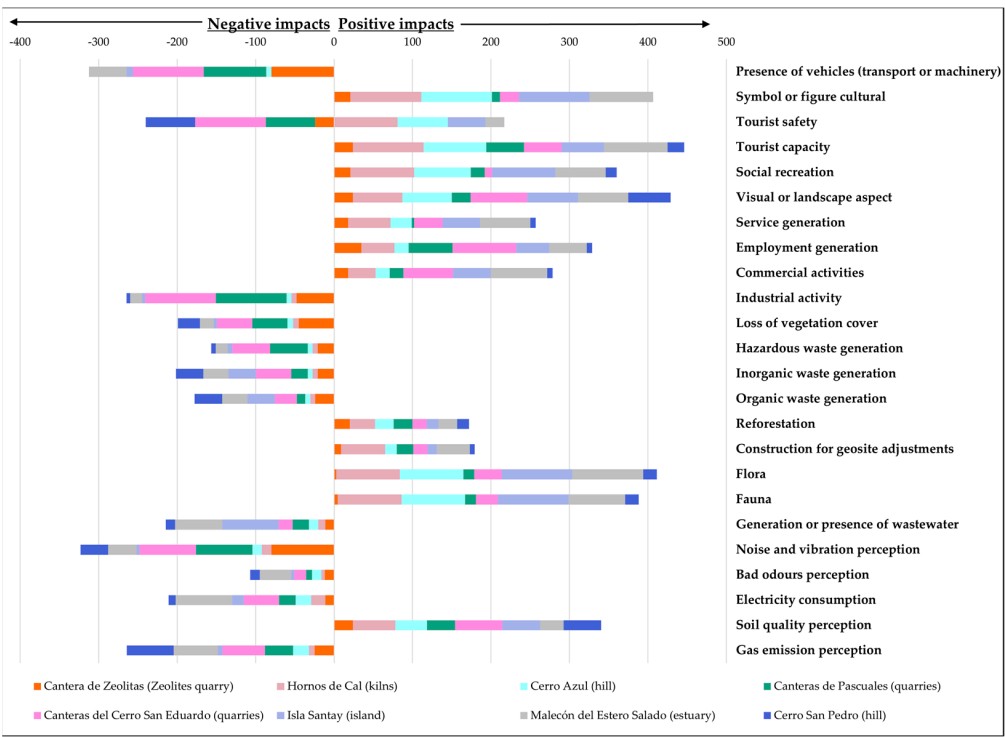

**Figure 5.** General schematization of the assessed environmental aspects.

The Hornos de Cal, Isla Santay, Estero Salado, and Cerro Azul geosites present a minimum number of negative environmental aspects, highlighting the elements of geo-biodiversity as a natural bridge to sustainability. Conversely, the quarries and Cerro San Pedro present a greater number of adverse environmental impacts, causing an imbalance in the natural ecosystem of the city.

The valuation of these sites determined that biodiversity (flora and fauna) is present in most of the areas of interest, linked to reforestation and adaptations of these places, except the quarries (including those of Cerro San Eduardo). In addition, geodiversity, due to its landscape load and cultural symbolism, allows for the generation of various services, jobs, and commercial activities. In terms of tourism, which allows for social recreation by tourists (except Cerro San Pedro and the quarries), this shows the key points that could enhance the interest and comfort of people (locals and visitors) because these places allow for geo-environmental conservation and sustainability.

On the other hand, vehicles (transport or heavy machinery) are present with greater force in the quarries than in the rest of the geosites. Therefore, aspects such as the perception of noise, vibrations, bad odors, and electricity consumption are found more frequently in most places, except for protected areas such as the Hornos de Cal in the Bosque Protector Cerro Blanco, which encourages conservation and care by visitors.

In general, this result determined the link between the most relevant environmental aspects with the daily activities of the city, such as industry, loss of vegetation cover (in

quarries), waste generation, and emission or perception of gases; this has an impact on the ecosystem and all of the sites of interest that were analyzed.

### 3.3. Most Significant Favorable Environmental Aspects

Figure 6 shows the magnitudes of positive evaluation (+) greater than 60 (indicated within the colored rectangles). In this context, values of more than 80 represent the strengths of the geosites, indicating those sites with the highest evaluation.

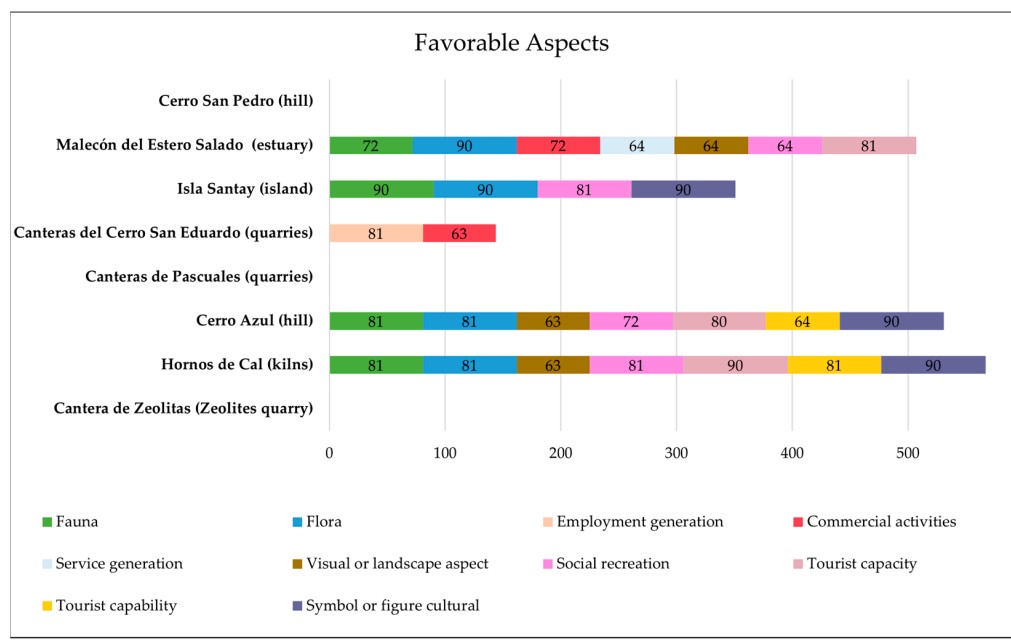

**Figure 6.** Favorable aspects in the valued sites of interest.

Most of the sites have aspects that highlight environmental interest. However, places like Cerro San Pedro and the quarries have few elements that benefit the sector. Except for the quarries on Cerro San Eduardo, where there is economic influence and employment generation that favor the industry, but not of the same magnitude in terms of environmental care.

Favorable aspects such as flora, fauna, social recreation, tourist security, employment generation, tourist carrying capacity, and the symbolic or cultural figure that the remaining sites have are strengths that benefit most of the evaluated sectors. These aspects promote the sustainability of the areas of interest to address the adverse impacts in each place.

Cerro Azul and the Hornos de Cal are natural reserves with a high degree of environmental benefit, essential for tourism in natural areas located on the city's outskirts. In this context, Santay Island has fewer aspects; however, it is a symbol or cultural figure of the country, where flora and fauna abound for visitors' research and social recreation. On the other hand, sectors within the city, such as Estero Salado, better promote commercial activities and the generation of services through various outdoor activities in a semi-natural environment.

### 3.4. Most Significant Unfavorable Environmental Aspects

Figure 7 shows the unfavorable aspects with extreme magnitudes between −60 and −100. The closer to −100, the greater the degree of affectation the site receives for each identified and valued adverse environmental impact. In addition, geosites such as Cerro Azul and Hornos de Cal are absent in the figure because they do not present significant unfavorable aspects (between −60 and −100).

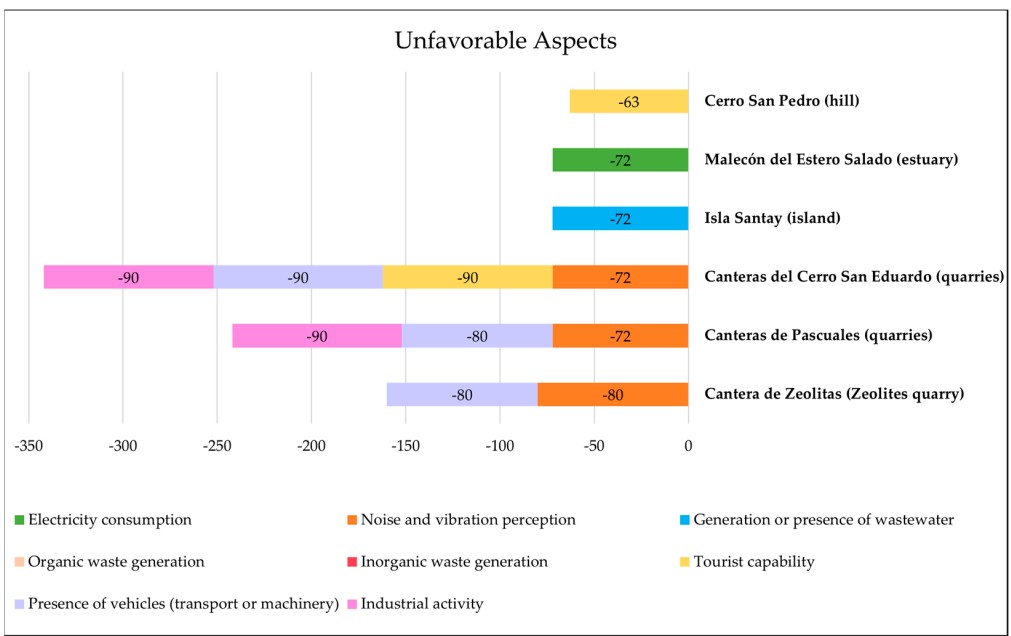

**Figure 7.** Unfavorable aspects in the valued sites of interest.

The quarries are the most affected due to the presence of vehicles and heavy machinery. Industrial activity in the sector increases environmental pollution due to the emission of noise, vibrations, and gases. Additionally, tourist security is not guaranteed, especially in the quarries on Cerro San Eduardo, where there is a high degree of affectation for future applications as visitor sites after the closure phase that all quarries must have. Cerro San Pedro and Canteras de Pascuales present a similar condition, mainly due to low security for visitors.

Sectors in the city's interior, such as Estero Salado, are mainly affected by high electricity consumption due to lighting in different areas, such as dining rooms, shops, sanitary batteries, and social recreation facilities (arcade games). However, in remote areas such as Santay Island, it is observed that there is a considerable impact due to the mismanagement of residual water and the waste generated in each sector (organic and inorganic).

### 3.5. Average Impact Value at Sites of Interest

Figure 8 represents the impact value in each place analyzed, based on the average value of all the aspects evaluated in this work. These values range from 0 to 100 in a positive (+) and negative (−) way, displayed at the bottom and where their magnitude refers to their impact or strengths.

Active quarries such as Pascuales and those located on Cerro San Eduardo have a greater negative environmental impact than the Zeolite quarry. However, because it is in a more isolated place and surrounded by dry vegetation (typical of the area), it would not affect the ecosystem's inhabitants (on a large scale). On the other hand, Cerro San Pedro, being an isolated place with little traffic, presents impacts related to the low level of security and low usage of its resources or characteristics.

Sectors in natural and ecological reserves, such as Hornos de Cal and Cerro Azul, present a better use of their natural resources, promoting a good relationship between nature and daily life in the city. Additionally, in sectors more embedded in the city, such as the Malecón del Estero Salado, there is a balance between anthropogenic activity and the environment. However, one of its main unfavorable aspects is electricity consumption. Unlike Santay Island, where many natural areas highlight the sector's geotourism value, it is affected by the low control of waste.

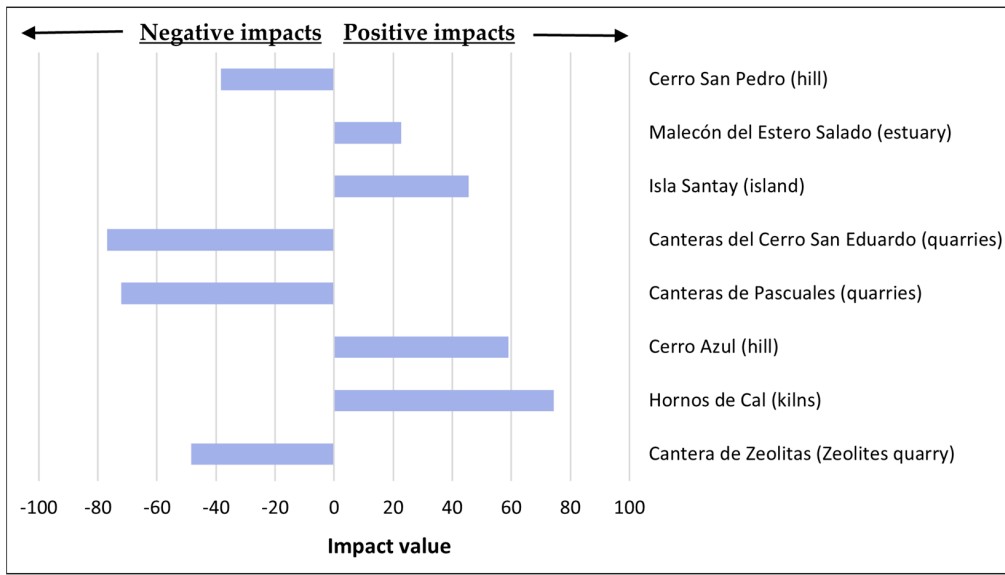

**Figure 8.** Average impact value of the rated sites.

All of this information highlights that these four geosites are the most suitable for generating georoutes and increasing tourism and improving the commercial and social activity of the city with important natural sites.

*3.6. Strategies for the Sustainability Proposal*

The evaluations determined several favorable aspects with multiple benefits for these sites. Therefore, ideas or proposals are proposed that mitigate the negative impacts identified, managing to promote or take advantage of the resources of each place to achieve their sustainability (see Table 4):

**Table 4.** Approach of strategies for sustainability.

| N.- | Strategies |
|---|---|
| #1 | Use or implement photovoltaic panels to reduce the energy consumption of the place, focused on isolated and natural areas with difficult adaptabilities, such as Santay Island. |
| #2 | Increase the shrub or tree vegetation to counteract the emission of gases and noise caused by the vehicular journey, thus improving the quality of life in the city. |
| #3 | Improve the wastewater system to minimize its effect due to poor distribution and disposal in sectors far from the city. |
| #4 | Take advantage of the natural environment (flora, fauna and the landscape) and promote geotourism as a geoconservation, reducing the impact on the ecosystem due to the various activities. |
| #5 | Properly manage the sites to encourage social recreation and the cultural symbol, represented in several of the natural and mixed sites (located in the anthropogenic contour of the city). |
| #6 | Improve the security provided to tourists in each sector to guarantee a good reception, increasing commercial activities, job creation, and services. |

## 4. Discussion

Implementing a matrix of geo-environmental evaluation of geosites in an urban area allows for a technical proposal for sustainable tourism development, contributing to the potential of geosites. In addition, it is a support and reference tool for decision-makers, and the key to the social and economic aspects of the local community.

The results of this study determined 24 impacts (12 were negative and 12 were positive). Similarly, four sites present unfavorable situations, while the other four have strengths (see Figure 8). The most significant negative aspects make it possible to propose corrective measures and preventive elements (Figure 7). Additionally, the strengths or favorable aspects allow for establishing the criteria to empower the geosites. The geosites have a natural diversity that characterizes them. However, there is an important connection with

the sector's cultural aspect, which magnifies tourism, as observed in the characteristics of each geosite in Table 2. These places have a cultural aspect that can be incorporated into the geographic context of the geosites through the existing link with the geology and natural geomorphology of the sector [144–146].

The application of a geo-environmental matrix focuses on a certain site's environmental and anthropogenic impacts (positive or negative). The matrix makes it possible to identify the strengths and weaknesses of each place, as in the work of Datta and Sarkar [147] in West Bengal (India), where they generated strategies to take advantage of or improve the benefits in situ and managing to encourage care and industry progress. Another example is the study by Marrosu and Balvis [65] in the rocky outcrops of Sardinia (Italy), where they determined the environmental value of the place and the magnitude of the impact of anthropic activities in the sector, which served to protect the geosites and manage the use of sports facilities without affecting the sustainability of the place.

The evaluation of geosites in urban areas is one of the environmental education initiatives used to improve the management of urban geoheritage, manifested in the work of Vegas and Díez-Herrero [148] in Segovia (Spain), together with the Geological and Mining Institute of Spain (IGME, acronym in Spanish), who focus on resilience issues within a combined urban and natural ecosystem. Furthermore, this initiative focuses on the promotion of geotourism and geopark proposals, such as the study by Özgeriş and Karahan [149] in Uzundere (the 11th city of Cittaslow, Turkey), where they evaluated the geopark's resources to promote the conservation of the natural environment (associated with the soil, the earth), which benefits all sectors and creates compliance with the SDGs.

This work focuses on the strengths of the geosites, mitigating the negative impacts and promoting tourism sustainability in the city. In sites far from the city, the following suggestions are proposed: (i) reduction of energy consumption through solar panels, as proposed by Lata-García et al. [150] on Santay Island; (ii) encourage geoconservation through geotourism as a means for geoeducation implemented in heritage sites and geoparks (as in India and Colombia) [151,152]; and (iii) improve the sewage systems and subsequent wastewater treatment in wetland areas surrounding the city, as proposed by Abrahams et al. [153] in a case study at Brookside Farm (UK), through a methodology of low consumption for the reuse of this water. On the other hand, in the interior of the city, it is proposed that sites: (iv) reduce the emission of gases, noise, and effects of heat through the increase of vegetation in wide areas suitable for its growth, verified in the Shimizu study et al. [154] (Japan) and Van Renterghem [155] (Belgium); (v) improve the city's natural and cultural sites to encourage social recreation, such as in the Chubusangaku National Park (Japan) [156]; and vi) ensure security to promote the resources and well-being of city sites, as suggested by Wang et al. [157] in a global vision of geoheritage sites.

The effects of vehicles are present in all of the geosites in a negative way due to the emission of gases and the generation of noise and vibrations. Tarcaya and Arenas [158], in a case study in Salta (Argentina), confirm that there are environmental impacts due to the emission of $CO_2$. One of the ways to counteract this is to improve the control of regulatory and environmental entities, given that sometimes there is a degree of contamination greater than that allowed, as evidenced in the city of Cuenca (Ecuador) [159]. However, in order to take advantage of the strengths of the sector and promote better comfort in the city, the inclusion of vegetation or green spaces should be chosen, as Selmi et al. [160] in a study in the city of Strasbourg (France), where it was evident that urban trees contributed to the elimination of air pollutants (7% of local emissions), for which they recommend improving the strategies for the planting of forest resources in urban structures.

The loss of vegetation cover is another negative aspect that is reflected to a great extent and is influenced by human intervention [161,162] due to activities related to urban development and mining exploitation, such as to the east of Buenos Aires (Argentina) [163]. In addition, there are cases such as Kalabsha, El-Sebaiya, and Medcom-Aswan (Egypt), where Ruban et al. [164] mention and recommend that quarries must have comprehensive management for the purpose of mining and tourism. Due to this, it is necessary to

consider and mitigate the anthropic effects, and even more so, to have revegetation and conservation activities for restoring the natural and cultural geoheritage, as mentioned by Damas et al. [165] in a study of the Gorria Quarry and Red Ereño (Spain). In addition, Prosser [166] highlights the link between the community and its geoheritage through quarries, using England as an example.

The generation of solid waste is present in some of the geosites and harms the environment. However, in the case of quarries, the waste can be used in producing or improving other products. As in the case of the brick industry in northeastern Italy, Coletti et al. [167] report that using trachyte residues from quarries enhances the properties of the new bricks (physical, mechanical, and durability). In addition, in road and building construction, Ramalakshmi and Bala Nagendra Babu [168] show that quarry dust is a good additive to improve soil compaction characteristics.

The generation or presence of wastewater is a negative aspect due to a load of heavy metals and metalloids, as reported by Pereda-Solis et al. [169] in a study in Durango (Mexico), demonstrating that there is zinc, lead, cadmium, and arsenic in the liver tissue of all waterfowl tested. For this case study, the place with the greatest impact is Santay Island, as corroborated by Zapata [170] through surveys of 56 families, who indicate that the systems and wastewater treatment are poor quality, which is also mentioned by Mora [171], due to the discharge of fecal coliforms into the river by the locality of the island and the city. However, this aspect can be controlled or mitigated so that it does not harm the ecosystem, as in arid cities in northern Chile, where they promote green area development through wastewater reuse [172].

The positive aspects make it possible to counteract the negative impacts, mainly those caused by human intervention and anthropogenic activity, as analyzed in the studies by Avelar et al. [173] and Fuertes-Gutiérrez et al. [174] (Região dos Lagos, (Rio de Janeiro—Brazil), and the paleontological sites of La Rioja (Spain), respectively), where sustainability problems are evident in the geosites. In the case of the present study, four geosites ( Santay Island, Horno de Cal, Cerro Azul, and Malecón del Estero Salado) are contemplated for a variety of activities for tourism development and sustainability. Similarly, sectors such as the Pascuales, Cerro San Eduardo, and Zeolites quarries can become historical sites or museums in situ due to the mining activity of yesteryear. As in the "Copper Trail" (Salzburg), where Ibetsberger and Steyrer [175] highlight its importance as a fundamental axis within a geopark ("Ore of the Alps Geopark"). In addition, there is the "Golden Triangle" (Egypt), where Ruban et al. [19] mention that the geosites included within this place can be part of the initiatives of the framework for the development of a Geopark.

This work carried out an overview of the geo-environmental state of the geosite through an environmental impact assessment (EIA), strictly derived from the Leopold matrix and the elements found in the supplementary material (Table S2). There are studies that, unlike this one, focus on addressing the impacts of the works on a specific natural site, such as Zaruma Urcu natural heritage site [176], where an environmental assessment identified that geotourism promotion works would have a significant environmental impact. Even so, the locality accepts and allows these changes to continue to improve the geosite. Another case study is that of Selmi L. et al. [177], in the Maltese islands, where they evaluated the environmental impact and risk of degradation of the geosites, demonstrating that there is negligence and ignorance on the part of the locality that causes mismanagement, which is why they provide information for adequate management and take advantage of their strengths. Tourist activities will always have an impact; therefore, it is important to know the tourist carrying capacity of geosites (especially those analyzed in this work,) since this allows for environmentally friendly management and development strategies for sustainable tourism [178,179]. For this reason, normal methods, such as IELIG or Brilha, consider the value of geosites within their evaluation [180].

## 5. Conclusions

The cause–effect geoenvironmental evaluation matrix can be replicated in sites of interest with similar characteristics, focusing on the aspects and effects caused by a project to be implemented. The use of this matrix allowed us to recognize the elements that affect the geosites of this study, such as (i) generation of noise and greenhouse gases caused by urban transport, (ii) generation of solid waste and residual water due to population growth, (iii) low levels of local and public security for tourists (in quarries), and (iv) energy consumption in tourist and natural areas. However, there are positive impacts related to the natural and tourist elements, such as (i) flora and fauna, (ii) generation of employment and services for social recreation, (iii) commercial activities, (iv) tourist capacity, (v) symbol or cultural figure and, and (vi) leisure and recreation spaces for tourists in natural areas close to the city. These elements are linked and integrated into the environment and geology, since some of them allow for the development of attractive activities for people linked to nature and the geological processes to encourage care, geoeducation, and sustainability in relation to the natural sites within a city. Currently, there are places that have been readjusted and recognized for tourism by the municipality, but this work seeks to include more places that highlight geology and geodiversity as a means for tourism.

The matrix allowed for the evaluation of eight places of tourist interest in Guayaquil and its surrounding areas. Cerro San Pedro and the quarries (Pascuales, Zeolites, and Cerro San Eduardo) have unfavorable conditions related to safety and industrial activity, in addition to presenting considerable air pollution (gases/noise) due to the presence of vehicles. Operating quarries do not promote tourism due to hazards and their negative effects identified in the study. However, they can be used as historical sites or museums after their closure phase, promoting geological and mining heritage to encourage geotourism, geoeducation, research, and social recreation in harmony with the environment. On the other hand, places like Cerro Azul, Malecón del Estero Salado, Santay Island, and Hornos de Cal present characteristics needed to promote sustainable geotourism since they are within protected areas and inserted in the city. Conservation and education strategies are key to improving the current state of geosites, given their nature and socioeconomic-political environment.

The previous results show that the impacts positively and/or negatively affect the sites. This forms the basis for management and development strategies. One way to benefit the sites and their inhabitants is by using tourism to impart geo-education to visitors, which promotes the evaluated geosites and improves the sector's economy. Through this, the work demonstrated that there are sites with positive impacts and that the implementation of tourism must consider the sustainability criteria that are focused on in this study, specifically from the environmental section, since it has several elements that strengthen the benefits to the local and economic sectors. Therefore, tourist use must have an adequate geo-education aspect that improves the connection between man and nature, provided that there are investment initiatives such as those proposed in Table 3.

The applied methodology is a tool that helps regulatory and government entities evaluate potential geosites and recognize those that need to improve their management. In this way, the proposal for a Geopark in Guayaquil and the surrounding areas is a challenge to sustainability. The limitations identified in this study are the number of geosites evaluated, since there are more places with characteristics of potential interest in the city, among those that stand out are Cerro Bellavista, Parque Lago, Balneario Puerto Hondo, and Bosque Protector Prosperina. Other limitations are the evaluation process and access to sites of interest. There are several qualitative methods to evaluate geosites; however, in some cases, they are semi-quantitative because they depend on the criteria and experience of the evaluators, as in this study, where they consider the average value assigned by each evaluator during the evaluation. On the other hand, specific permits are needed for access in some places because they are quarries in operation or they are private places.

**Supplementary Materials:** The following supporting information can be downloaded at: https: //www.mdpi.com/article/10.3390/heritage6030153/s1, Table S1: Experts involved in the cause-effect geo-environmental assessment matrix.; Table S2: Description of the aspects identified in situ.

**Author Contributions:** Conceptualization, F.M.-C., B.A.-M., P.C.-M., F.T.-M., B.M.-S., L.S.-N. and G.H.-F.; methodology, F.M.-C., B.A.-M., P.C.-M., F.T.-M., B.M.-S., L.S.-N. and G.H.-F.; software, B.A.-M., F.T.-M. and L.S.-N.; validation, F.M.-C., B.A.-M., P.C.-M., F.T.-M., B.M.-S., L.S.-N. and G.H.-F.; formal analysis, F.M.-C., B.A.-M., P.C.-M., F.T.-M., B.M.-S., L.S.-N. and G.H.-F.; investigation, F.M.-C., B.A.-M., P.C.-M., F.T.-M., B.M.-S., L.S.-N. and G.H.-F.; resources, F.M.-C., B.A.-M., P.C.-M. and F.T.-M.; data curation, B.A.-M. and F.T.-M.; writing—original draft preparation, F.M.-C., B.A.-M., P.C.-M., F.T.-M., B.M.-S., L.S.-N. and G.H.-F.; writing—review and editing, F.M.-C., B.A.-M., P.C.-M., F.T.-M., B.M.-S., L.S.-N. and G.H.-F.; visualization, B.A.-M.; supervision, F.M.-C., P.C.-M., B.M.-S., L.S.-N. and G.H.-F.; project administration, F.M.-C., B.A.-M. and P.C.-M.; funding acquisition, F.M.-C. and P.C.-M. All authors have read and agreed to the published version of the manuscript.

**Funding:** This research received no external funding.

**Data Availability Statement:** Not applicable.

**Acknowledgments:** This research study was possible with the valuable contribution of the "Registry of geological and mining heritage and its impact on the defense and preservation of geodiversity in Ecuador" academic research project at ESPOL University CIPAT-01-2018.

**Conflicts of Interest:** The authors declare no conflict of interest.

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
