# Peer review of "Geo-Environmental Assessment of Tourist Development and Its Impact on Sustainability"

_heritage, doi:10.3390/heritage6030153_

Round 1

Reviewer 1 Report

The manuscript focusses on environmental assessment of sites used for, or that can be used for tourism. This focus is valuable and deserves attention. The assessment follows logics from Environmental Impact Assessments (EIA) utilising the ideas of Leopolds matrice method. The manuscript is, however, not clear. The reason for this is that the assessment partly is very focused on details and partly of a very general nature. A large part of this assessment is linked to general environmental issues such as pollution, climate, noise etc. It is not clear if the elements of the assessment is linked to present status and present use or future possible situations given different scenarios of use. It is not clear if touristic use, and what sort of investments needed for this use that increases or decreases the present impact. The text is general, and lack of specifications linked to the case study makes it difficult to follow, both with respect of relevance and effect.

The text is very concerned about local economic development and welfare, and this seems reasonable and is perfectly valid. It is, however, not clear when, and how this perspective can be linked to sustainability. When do local tourist development increase or decrease human footprint, and how much – and in relation to alternative activities.

The title of the manuscript is not clear. If I understand it correctly it could be: “Geo-environmental assessment of tourist development to improve sustainability”.

The manuscript focuses on geodiversity, geological heritage and geoconservation. The relation with these issues is, however, weak. I suggest including a text in the introduction that describes the geography and geology of the area, how the local geodiversity is reflected in defined geoheritage, the vulnerability and resilience of the area with respect to geoheritage and the threat to these values from a) urban development and b) tourist activities. Here it will be room to specify what sort of tourist development that can mitigate such problems and still work as a local economic resource. As I can see it it is three different geodiversity profiles present in the different sites described: a) Natural fluvial/intertidal processes, Natural outcrops linked with landscape features and 3) quarries. They could well be better described within the focus mentioned above. It is not clear if the touristic focus on these sites is general or linked to geology. The text indicates ecology, landscape, and leisure as main focus, but do not specify how geology is or can be integrated in their use and what sort of effect that will introduce. The exception is the quarries. Here, however, it is a need to specify the present status better. How can these be used for geotourism. What sort of geoheritage do they represent. Is it possible to use them when the quarry still is in operation and if they are revegetated after operation, will geoheritage still be visible so it can be used for research, education and geotourism? The tables indicating their approval percentage etc. is not easy to interpret as the methods used is only referenced and as far as I can assess includes a multitude of factors and do not isolate factors vital for the assessment of let’s say scientific value, vulnerability, and other factors vital for their use and conservation.

The terms geodiversity, geological heritage and geoconservation are defined very and too narrow. I recommend definitions more widely used, for example those suggested by Murray Gray that already ais referenced. It is not obvious that geotourism and specifically geopark development can be defined as geoconservation. Geoconservation is an activity that should be developed within these activities, and it does not come as a formal and obvious result of these activities. I recommend the manual for geoconservation in protected areas published by IUCN. It would be interesting to know a bit more of the management system in use in the area, as management is a vital part of geoconservation.

Author Response

We greatly appreciate your comments and observations, allowing us to improve the article based on your review. According to what was sent by the journal, five paragraphs describe their comments and suggestions. These were broken down to attend to each one and meet this objective. Below are the changes made: 

Reviewer 2 Report

I enjoyed reading this research. The idea is excellent, the literature review, methodology, results, discussion and conclusion are also excellent. Congratulations to the authors. Good luck!

Author Response

We are grateful for your words and the positive message given to our work.

Reviewer 3 Report

The article is interesting and deals with an important subject and certainly is a basis for a future geopark candidature to the UNESCO of the different geosites presented. Even so, I have a few suggestions to improve its quality.

In the introduction section, since it includes a definition of the geodiversity concept and the date when the concept arose in the science community, I would recommend a similar inclusion concerning the concept of geopark.

The cause-effect matrix to assess the different geosites is an interesting tool. However, nothing is said about: how the process of evaluation was implemented; how many scientists participated in the assessment and their academic profile; how possible discrepancies in the team during the assessment were overcome?

The discussion section is interesting, since it includes both the positive and negative aspects of the different geosites, as well as possible ways to mitigate the negative aspects. Even so, the connections between the natural features of the geosites and their cultural importance could be clearly approached.

Finally, I would suggest the inclusion of other limitations of the study in the conclusions section, more related to the assessment process of the geosites that were analysed, and not only the reference that more places could be object of assessment.

Author Response

We greatly appreciate your comments and observations, allowing us to improve the article based on your review. Below are the changes made: 

Reviewer 4 Report

this paper is about The evaluation of geosites as an essential part of conserving the geodiversity and biodiversity of the ecosystem. Through the chosen case study, the authors address the issue in an orderly manner. The structure of the article is solid (theoretical premises, objectives and methods, analysis and discussion of the results). The scientific soundness is adequate. The cartographic and graphic equipment is remarkable. The bibliography is up-to-date and extensive.

Author Response

We are grateful for your observations and review of our work.

Round 2

Reviewer 1 Report

The manuscript has been improved, though  it is elements that still could be adjusted, see comments below:

It still lack a discussion about the impact of tourism on the geoheritage value as such. in what respect the geological elements ar vulnurable for tourist activities, have resilience or are impacted positively or negatively by the development. If this is not done it should be defined clearly that t5his is outside the scope and be more clear that this is a general EIA of tourist development strictly based on the Leopold matrix and elements found in supplementary material. It should then be stated that such assessments (geological) already has been done and included in the tables of suitability for defining geosites (with references). This implies that geosites are not valued on their scientific value alone, but selected for their ability to cope with increased tourism. This again implies that it is not an ionvestigation of geodiversity values as such, but a selection of sites for use. A clarification to this is much needed.

Suplementary material (part 2) is very important and should be very clearly referenced or even considered included as a table.

Abstract: I guess Spanish text should not be here: La geodiversidad, tal como lo menciona Gray M. [5], es el equivalente abiótico de 68 la biodiversidad y columna vertebral del geopatrimonio y geoconservación,

Line 70-76: To establish a geopark geosites of international value is needed. Geotourism is in the centre of its aim. Perhaps the text is a bit too careful? Sector - Do you mean geograpohical (area, district) or economic sector i.e tourism?

line 79: "as a tool" as alternative to "as a means"

Line 92: "achieving  a record or categorization of megadiverse places that require local participation" - a geosite do not have to be megadiversi or diverse for that matter. Many value criteria exsist.  Local participation is also a matter that can vary from site to site. Of course management improves with local partisipation.

Line 136: "in which geodiversity linked" as alternative for "in which a geodiversity linked"

Line 143: "There is a unique local geodiversity difining the geoheritage of the area linked to natural,-, ancient and historical areas framed by the natural environment, urban development and human activities of yesteryear." Rather than:  "There is a unique local geodiversity as part of the represented geoheritage, such as natural, ancient and historical sectors framed by the environment, urban development and activities of yesteryear. (if I interpret the message correctly)

line 171: "from the same article" rather than "of the same article" I believe.

Figure captions: A short explanation of the numbers would be nice.

Line 460: "sites" rather than "site". Please clarify the sentence. "Currently, there are places readjusted and recognized for tourism by the municipality, but this work seeks to include more places that highlight geology and geodiversity as a means for tourism." Is it relevamnt? Does these sites represent geoconservation to a higher degree than the studied sites. Should thy undergo the same environmental assessment as described??

Line 476-486: Need clarification. Try with simpler sentences. excample: "The previous results determined that the elements or aspects affect the sites of interest positively and negatively, through which strategies are developed based on the strengths to reduce their weaknesses." What are elements and aspects? Is this correctly understood: "The previous results show that the impacts affect the sites positively and/or negatively. This form the basis for management and development strategies." ?

Line 497: What do the sentence "In addition, some need specific permits for access to the places due to their origin/activity." mean and is it relevant for the conclution?

Author Response

Dear reviewer, the authors appreciate your comments on the article, which will undoubtedly improve it.

We have responded to each comment and attached a document for your review.

We appreciate your help.
